# Antioxidant Activities of Dihydromyricetin Derivatives with Different Acyl Donor Chain Lengths Synthetized by Lipozyme TL IM

**DOI:** 10.3390/foods12101986

**Published:** 2023-05-14

**Authors:** Baoshuang Du, Shan Wang, Song Zhu, Yue Li, Dejian Huang, Shangwei Chen

**Affiliations:** 1State Key Laboratory of Food Science and Technology, Jiangnan University, Wuxi 214122, China; zhu_song2004@163.com (B.D.); zhusong@jiangnan.edu.cn (S.Z.); chenshangwei@jiangnan.edu.cn (S.C.); 2International Joint Laboratory on Food Safety, Jiangnan University, Wuxi 214122, China; 3School of Food Science and Technology, Jiangnan University, Wuxi 214122, China; 7210112052@stu.jiangnan.edu.cn; 4Department of Food Science and Technology, National University of Singapore, Singapore 117543, Singapore; fsthdj@nus.edu.sg

**Keywords:** dihydromyricetin, acylated derivatives, antioxidant activity, lipophilicity, cellular antioxidant activity

## Abstract

Dihydromyricetin (DHM) is a phytochemical with multiple bioactivities. However, its poor liposolubility limits its application in the field. In this study, DHM was acylated with different fatty acid vinyl esters to improve its lipophilicity, and five DHM acylated derivatives with different carbon chain lengths (C2-DHM, C4-DHM, C6-DHM, C8-DHM, and C12-DHM) and different lipophilicity were synthesized. The relationship between the lipophilicity and antioxidant activities of DHM and its derivatives was evaluated with oil and emulsion models using chemical and cellular antioxidant activity (CAA) tests. The capacity of DHM derivatives to scavenge 1,1-diphenyl-2-picrylhydrazyl radical (DPPH•) and 2,2’-azino-bis(3-ethylbenzothiazoline-6-sulfonic acid) radical (ABTS+•) was similar to that of DHM, except for C12-DHM. The antioxidant activity of DHM derivatives was lower than that of DHM in sunflower oil, while C4-DHM exhibited better antioxidant capacity in oil-in-water emulsion. In CAA tests, C8-DHM (median effective dose (EC_50_) 35.14 μmol/L) exhibited better antioxidant activity than that of DHM (EC_50_: 226.26 μmol/L). The results showed that in different antioxidant models, DHM derivatives with different lipophilicity had various antioxidant activities, which has guiding significance for the use of DHM and its derivatives.

## 1. Introduction

Dihydromyricetin (DHM), called ampelopsin, is the main dihydroflavonol flavonoid compound in vine tea and widely exists in Garcinia, Vitis, and Myricaceae plants [1]. It has a variety of biological functions, including scavenging free radicals, antioxidation, antitumor, anti-inflammatory, inhibiting hepatocyte deterioration, accelerating liver cell proliferation, and reducing liver injury [2,3,4]. DHM has six active hydroxyl groups in its molecular structure, which makes it exhibit strong antioxidant activity. However, it also makes DHM have strong polarity, resulting in poor solubility in lipid systems and poor penetration through lipophilic membrane barriers, which limits the application of DHM in lipid and biological systems. At present, DHM can be modified by acylation reactions to improve its lipid solubility. Enzymatic acylation is the most common acylation method, which has mild reaction conditions and high selectivity. In fact, there are many studies on the acylation modification of natural antioxidants, including chlorogenic acid [5], (−)-epigallocatechin gallate (EGCG) [6], quercetin [7,8], and resveratrol [9]. However, there are few reports on the acylation modification of DHM and the biological activity of DHM acylated derivatives.

Lipid oxidation may cause corruption and waste of products and poses a certain threat to human health, which is an important issue in food industry [10]. Therefore, the addition of antioxidants to delay lipid oxidation has also become a research hotspot. At present, acylation-modified natural antioxidants have been widely studied in bulk oil and emulsion systems. Some studies have shown that the polarity of antioxidants affects their antioxidant effect in emulsion and bulk oil systems. According to the polarity paradox [11,12], lipophilic antioxidants have a good effect in emulsions, while the effect of hydrophilic antioxidants in bulk oil systems is good. However, not all antioxidants follow the polarity paradox. Oh and Shahidi [13] showed that the antioxidant effect of resveratrol acylation products in a bulk oil system was better than that of resveratrol. The research of Laguerre et al. [14] also reported that the antioxidant effect of the compound in oil-in-water emulsion (O/W emulsion) is not simply positively correlated with its lipid solubility. There is a cut-off effect on the antioxidant activity of some antioxidant acylation products in lipid soluble systems [14,15,16,17]. Meireles et al. [16] found that with the extension of the carbon chain, the antioxidant activities of chlorogenic acid and its alkyl esters in O/W emulsion first increased and then decreased. However, the antioxidant activity of resveratrol and its acylated products in the O/W emulsion did not show a cut-off effect, and the antioxidant effect of acylated products was lower than that of resveratrol [13]. Therefore, the polarity paradox and cut-off effect are specific to antioxidants, and the best chain lengths of different antioxidants in emulsion and bulk oil systems are different. It is of great significance to study the antioxidant effect of DHM and its derivatives in emulsion and bulk oil systems to find the best chain length.

In addition, research on the acylated derivatives of antioxidants mainly adopts in vitro chemical tests, such as 1,1-diphenyl-2-picrylhydrazyl (DPPH) and 2,2’-azinobis (3-ethylbenzothiazoline-6-sulfonic acid) (ABTS) free radical scavenging. These methods are fast, simple, and sensitive, but the reaction medium is relatively singular, which does not reflect the process of antioxidant reactions in vivo, and many substances with strong antioxidant activity evaluated in vitro may have strong cytotoxicity in vivo, which limits their application [18]. The cell method is cost-effective, rapid, and takes more factors into account such as cytotoxicity, bioavailability, and stability. The determination results are more accurate and effective [19].

In this study, DHM derivatives with various carbon chain lengths were prepared with Lipozyme TL IM as a catalyst, and their lipophilicity, as well as their antioxidant activity in different models (chemical tests, lipid-based oil, emulsion food systems, and cell models), was studied to explore the relationship between lipophilicity and antioxidant activity of DHM and its derivatives.

## 2. Materials and Methods

### 2.1. Materials

DHM, purity >98%, was purchased from Bomei Biotechnology Ltd. (Hefei, Anhui, China). Lipozyme TL IM (250 U/g) was obtained from Amano Enzyme Inc. (Nagoya, Aichi, Japan). Vinyl butyrate, vinyl hexanoate, vinyl octanoate, and vinyl laurate were obtained from Tokyo Chemical Industry Ltd. (Tokyo, Japan). ABTS and DPPH were purchased from Shanghai McLean Biochemical Technology Co., Ltd. (Shanghai, China). 2,2’-azobis (2-amidinopropane), dihydrochloride (ABAP), and 2’,7’-dichlorofluorescein diacetate (DCFH-DA) were obtained from Sigma–Aldrich (Shanghai) Trading Co., Ltd. (Shanghai, China). Thirty percent H_2_O_2_, L-ascorbic acid (Vc), *tert*-butylhydroquinone (TBHQ), trichloroacetic acid, 2-thiobarbituric acid, ammonium thiocyanate, *n*-butanol, isooctane, isopropanol, chloroform, glacial acetic acid, sodium thiosulfate, Tween 20, methyl *tert*-butyl ether (MTBE), and 1,1,3,3-tetraethoxypropane were provided by Shanghai Sinopharm Group Chemical Reagent Ltd. (Shanghai, China). An MTT assay kit was obtained from Nanjing Jiancheng Bioengineering Institute (Nanjing, Jiangsu, China). RPMI 1640 culture medium, dimethyl sulfoxide (DMSO), fetal bovine serum (FBS), phosphate buffer solution (PBS), 0.25% trypsin, penicillin, and streptomycin were purchased from Thermo Fisher Scientific Inc. (Waltham, MA, USA).

### 2.2. Preparation of Acylated Derivatives of DHM

The preparation of DHM derivatives involved the use of a method reported by Zhu et al. [20]. DHM (0.18 mmol), five kinds of acyl donors (3.6 mmol), Lipozyme TL IM (0.4 U/mg DHM), and MTBE (10 mL), which were mixed in the reaction bottle, and the mixture was placed on a magnetic stirrer for reaction at 200 rpm and 50 °C for 72 h. After the reaction, the mixture was separated and purified by Waters 2545 prep-HPLC and then lyophilized.

After the structural identification of the reaction products by LC–MS and nuclear magnetic resonance spectroscopy, it was determined that the substitution site was at the 3-OH of DHM [20]. Therefore, five DHM acylation derivatives were synthesized, and their purity was >90%, including 3-*O*-acetyl-DHM (C2-DHM), 3-*O*-butyryl-DHM (C4-DHM), 3-*O*-hexanoyl-DHM (C6-DHM), 3-*O*-octanoyl-DHM (C8-DHM), and 3-*O*-lauroyl-DHM (C12-DHM). The structural formulas of DHM derivatives are in Appendix A.

### 2.3. Determination of the Partition Coefficient of Lipids and Water (log P)

The lipophilicity of DHM and its derivatives was evaluated using log P, which was determined by a previous method [19] with several modifications. Two milligrams of sample were dissolved in *n*-octanol solution (4 mL) saturated by water and analyzed using HPLC at 298 nm (A_0_). The *n*-octanol solution (1 mL) saturated by water containing the sample was mixed with water solution (1 mL) saturated by *n*-octanoland and then shaken using a constant-temperature water bath oscillator (Tianjin, China) at 30 °C for 24 h at 150 rpm. The ultraviolet absorbance of the upper solution was analyzed at 298 nm (A_x_). The calculation equation of log P is as follows:(1)$${\log\;P=\log\;A}_{x}{\;/\;(A}_{0\;}-{\;A}_{x})$$

### 2.4. Antioxidant Activity

#### 2.4.1. DPPH Radical Scavenging Ability

The method used to determine the abilities of DHM and its acylated derivatives was a previous method with some modifications [21]. Two milliliters of DPPH (0.065 mg/mL) and 2 mL of the sample (3, 5, 10, 15, and 20 μg/mL) were added to a centrifuge tube and described at 517 nm after 30 min of reaction in a dark environment. Vc (a water-soluble antioxidant) and TBHQ (a lipid soluble antioxidant) were used as controls.
(2)$$\mathrm{DPPH}\;\mathrm{radical}\;\mathrm{scavenging}\;\mathrm{activity}\;(\%)\;=(1-\frac{A_{1}{\;-\;A}_{2}}{A_{0}})\;\times \;100$$
where A_0_, A_1_, and A_2_ represent the absorbance values of the DPPH-ethanol, DPPH-sample, and sample-ethanol mixtures, respectively.

#### 2.4.2. ABTS Radical Scavenging Ability

The ability of DHM and its acylated derivatives to scavenge ABTS radicals was evaluated using a previous method [19] with some modification. Samples (0.1 mL, 0.01, 0.02, 0.04, 0.06, and 0.08 mg/mL) were added to 3.9 mL of ABTS solution and measured at 734 nm after incubation at 30 °C for 6 min. Vc (a water-soluble antioxidant) and TBHQ (a lipid soluble antioxidant) were used as controls.
(3)$$\mathrm{ABTS}\;\mathrm{radical}\;\mathrm{scavenging}\;\mathrm{activity}\;(\%)\;=(1-\frac{A_{1}{\;-\;A}_{2}}{A_{0}})\;\times \;100$$
where A_0_, A_1_, and A_2_ represent the absorbance values of the ABTS-ethanol, ABTS-samples, and samples-ethanol mixtures, respectively.

### 2.5. Antioxidant Activity in Sunflower Oil

The antioxidant activities of TBHQ used as control, DHM, and its derivatives in sunflower oil were compared. Samples (200 mg/kg oil) were added to sunflower oil and mixed evenly, and they were placed in an oven at 50 °C to accelerate oxidation. The peroxide value (POV) was determined every 3 days. The POV was measured using a method of Zhu et al. [6]. with slight modification. Oil sample (2–3 g) was added to a chloroform-glacial acetic acid mixture (30 mL, 2:3, *v*/*v*). Then, saturated potassium iodide solution (1 mL) was added. The mixture was shaken gently for 0.5 min and then stewed for 3 min in the dark. Subsequently, 100 mL of water was added and shaken and titrated with sodium thiosulfate standard solution to light yellow immediately. Then, starch indicator (1 mL) was added. The mixture was titrated continuously and shaken strongly until the blue color of the solution disappeared. The POV was calculated using the following equation:(4)$$\mathrm{POV}=\frac{{(V\;-\;V}_{0})\;\times \;c\;\times \;0.1269}{m}\;\times \;100$$
where V and V_0_ are the volume of sodium thiosulfate standard solution consumed by the sample and blank test (mL), respectively; c is the concentration of sodium thiosulfate standard solution (mol/L); m is the weight of oil sample (g); and 0.1269 is the mass of iodine equivalent to 1.00 mL sodium thiosulfate standard solution.

Another 3 g of oil samples with different antioxidants were taken to measure induction time using Rancimat (743 model, Metrohm, Switzerland). The air flow was 20 L/h, and the temperature was 120 °C. The induction time was determined according to the time when the recording curve reached the inflection point.

### 2.6. Antioxidant Activity in the O/W Emulsion

#### 2.6.1. Emulsion Preparation

Tween 20 (1% *w*/*v*) was added to phosphate buffer (10 mmol/L, pH 6.5), and 10% sunflower oil was mixed with 90% aqueous phase. The mixture was dispersed for 3 min at high speed at 15,000 rpm and then homogenized two times under 40 MPa. DHM and its acylated derivatives were added into emulsions (the sample concentration was 0.35 mmol/L). Finally, emulsions were placed in an oven at 45 °C to accelerate oxidation.

#### 2.6.2. Determination of Peroxide Value

The peroxide value was determined using the period method [22] with several modifications. Then, 0.2 mL of emulsion and 1.3 mL of isooctane/isopropanol (2:1, *v*/*v*) were mixed, vortexed for 1 min, and centrifuged at 6102× *g* for 5 min. Then, the upper solution (200 μL) was mixed with Fe^2+^ solution (0.072 mol/L, 20 μL) and ammonium thiocyanate solution (3.94 mol/L, 20 μL), and a methanol/n-butanol mixture (2:1, *v*/*v*) was added to the volume to 3 mL. The absorbance was measured at 510 nm after incubation in the dark for 20 min. The peroxide value was calculated by a standard curve of the Fe^3+^ standard solution.

#### 2.6.3. Determination of TBARS Value

The TBARS value of emulsion was determined by method [17] with some modifications. One milliliter of emulsion was mixed with 2 mL TBARS reagent (15 g of trichloroacetic acid and 0.375 g of 2-thiobarbituric acid were dissolved in 100 mL of water, and 2.1 mL of HCl was added) and heated in a boiling water bath for 15 min. The samples were cooled rapidly and then centrifuged at 6102× *g* for 10 min. The absorbance was measured at 532 nm. The TBARS value was calculated by a standard curve of 1,1,3,3-tetraethoxypropane.

#### 2.6.4. Determination of Antioxidant Partitioning in the O/W Emulsion

The physical location of DHM and its derivatives in the emulsions was determined using the procedure [23]. One milliliter of the emulsion-added sample was centrifuged at 12,205× *g* for 2 h. The lower water phase was collected carefully and analyzed by an Agilent 1260 HPLC system (Santa Clara, CA, USA).

### 2.7. Cellular Antioxidant Activity (CAA)

#### 2.7.1. Cell Culture

Human normal hepatocytes (L-02 cells, passage number 4) were purchased from Cell Bank of Shanghai Institutes for Biological Sciences, Chinese Academy of Sciences (Shanghai, China) and cultured in RPMI 1640 medium (Invitrogen, Carlsbad, CA, USA) supplemented with 10% FBS and 1% penicillin-streptomycin double antibody and were cultured at 37 °C in 5% CO_2_.

#### 2.7.2. Cytotoxicity

L-02 cells were inoculated in 96-well plates (7 × 10^3^/well, 100 μL) and incubated for 24 h. Then, the culture medium was removed, the cells were washed with PBS, and 100 μL of sample (0, 30, 60, 120, 240, and 480 μmol/L) was added. After incubation at 37 °C for 24 h, 50 μL of MTT solution (5 mg/mL) was added to each well and incubated for 4 h. Finally, the medium was removed and DMSO (150 μL) was added to dissolve formazan by shaking for 10 min. The absorbance was determined at 570 nm. Control wells and blank wells were set. The control wells were cells without sample treatment, and the blank wells were culture medium without samples and cells. Survival rates above 90% were considered nontoxic.
(5)$$\mathrm{Cell}\;\mathrm{survival}\;\mathrm{rate}\;(\%)=\frac{A_{\mathrm{sample}}-{\;A}_{\mathrm{blank}}}{A_{\mathrm{control}}-{\;A}_{\mathrm{blank}}}\;\times \;100$$

#### 2.7.3. Cellular Antioxidant Activity (CAA) Assay

The CAA assay was made using the method of Wolfe and Liu [24] with modifications. L-02 cells were inoculated in black 96-well plates (6 × 10^4^ /well, 100 μL) and incubated for 24 h. After removing culture medium, and the cells were washed with PBS and treated with the samples and DCFH-DA (25 μmol/L) for 1 h. Then, the cells were divided into two groups. One group was washed with PBS, and ABAP (600 μmol/L, 100 μL) was added to each well. Another group was only treated with adding 100 μL of ABAP (600 μmol/L) without washing. The black 96-well plates were measured by a Thermo Scientific Fluoroskan Ascent FL (Thermo Fisher Scientific, Waltham, MA, USA) every 5 min for 1 h at an emission wavelength of 538 nm and an excitation wavelength of 485 nm. Three control wells and one blank well were set. The control well was made up of the cells treated with ABAP but without samples, and the blank well was made up of the cells treated without ABAP and samples. Quercetin was used as the positive control.
(6)$$\mathrm{CAA}\;\mathrm{unit}=\left(1\;-\;\frac{\int \mathrm{SA}}{\int \mathrm{CA}}\right)\;\times \;100$$
where ∫SA represents the area integral under the sample fluorescence value *versus* time curve and ∫CA represents the area integral under the control fluorescence value *versus* time curve.

### 2.8. Statistical Analysis

All experimental data were analyzed by Origin 2018 (OriginLab Corp, Northampton, MA, USA) and IBM SPSS Statistics 22 (IBM Corp, Armonk, NY, USA) using the methods of nonlinear regression and Duncan’s multiple-range test. The statistical significance threshold was set to *p* < 0.05.

## 3. Results and Discussion

### 3.1. Evaluation of Antioxidant Activity

Previous experiments showed that the conversion rate of fatty acid vinyl ester with longer carbon chain and DHM reaction was relatively low, and after 72 h of reaction, the conversion rate of vinyl palmitate (C16) was only 45.9%. In addition, the reaction product of fatty acid with longer carbon chain has weaker polarity and lower solubility in methanol-water system, which leads to the precipitation of product during purification using Prep-HPLC, resulting in pipeline blockage and difficulty in preparation. Antioxidant experiments show that the activity of C12-DHM is not the best; therefore, C12 was selected as the largest carbon chain length to prepare DHM acylated derivatives in this study.

Acylation improved the lipophilicity of DHM, and the log P value also showed that with the extension of the substituted alkyl chain, the log *p* value increased and the lipophilicity increased (DHM, 0.81; C2-DHM, 1.32; C4-DHM, 1.50; C6-DHM, 1.67; C8-DHM, 1.73; and C12-DHM, 1.80). At the same time, the antioxidant activity of polyphenols is related to their polyhydroxy structure. The acylation reaction occurs when an acyl group replaces a H atom of hydroxyl group in the DHM structure, which affects its antioxidant activity to a certain extent.

The DPPH radical scavenging assay used to evaluate the antioxidant activity of antioxidant was based on this assumption that the antioxidant capacity of substance was equal to its electron-donating capacity or reducing capacity. As shown in Figure 1A, the abilities of Vc and TBHQ to scavenge DPPH radicals were higher than those of DHM and its derivatives. In the concentration range of 3–20 μg/mL, the scavenging abilities of DHM and its acylated derivatives to DPPH radicals increased with increasing concentration. When the concentration of sample was 20 μg/mL, the scavenging rates of DPPH free radicals by DHM, C2-DHM, C4-DHM, C6-DHM, C8-DHM, and C12-DHM were 93.37 ± 1.83%, 92.58 ± 1.33%, 92.70 ± 0.79%, 82.53 ± 1.98%, 74.14 ± 0.40%, and 29.62 ± 4.88%, respectively. The DPPH radicals scavenging rate of C12-DHM showed the maximum decline (*p* < 0.05). The IC_50_ values of the DPPH radical scavenging rates were DHM ≈ C2-DHM ≈ C4-DHM ≈ C6-DHM < C8-DHM < C12-DHM (*p* < 0.05). DHM derivatives exhibited the scavenging abilities of the DPPH radical that was not significantly different from DHM (*p* > 0.05), while the IC_50_ value of DPPH radical scavenging increased significantly when the alkyl chain increased to C12 (Table 1), which was approximately 4.76 times that of DHM. The acylation reaction replaced an active hydroxyl group in the DHM molecule, which affected its antioxidant capacity [19]. In addition, the increase in hydrophobicities of DHM derivatives will affect their distribution in polar media and then affect the abilities to scavenge free radicals [25]. Some studies have shown that the scavenging capacities of resveratrol alkyl esters on DPPH radicals are lower than those of resveratrol, and the scavenging capacities of alkyl esters with different carbon chains show the trend of first decreasing and then increasing [13]. The antioxidant activities of acylated derivatives are affected by the substitution positions, the lipophilic degrees, and the reaction media [26,27].

ABTS radical scavenging assay is also a common method used to measure the total antioxidant ability of substance. As shown in Figure 1B, in the concentration range of 0.01–0.08 mg/mL, the scavenging capacity of each sample for ABTS radicals increased with increasing concentration and showed a linear relationship. At 0.08 mg/mL, the scavenging rates of DHM and its derivatives to the ABTS radical followed the order of DHM > C2-DHM ≈ C4-DHM > C6-DHM ≈ C8-DHM > C12-DHM (*p* < 0.05), and from Table 1, when the carbon chain increased from C0 to C8, the IC_50_ values were not significantly changed (*p* > 0.05), and only C12-DHM increased significantly (*p* < 0.05). In general, the abilities of DHM and its acylated derivatives to scavenge ABTS radicals gradually decreased with increasing lipophilicity. A study suggested that ABTS radicals were free radicals with weak lipophilicity and possessed strong absorbance at 734 nm [28]. The existence of antioxidants would prevent generating ABTS free radicals. However, the log *p* values of DHM derivatives and the lipophilicity of DHM derivatives were improved, which means the affinity of DHM and its derivatives with ABTS free radicals decreased with increasing lipophilicity, resulting in a decrease in scavenging ability. DHM and its derivatives exhibited better antioxidant capacities in DPPH than those in the ABTS assays. This is because the DPPH assay is based on a radical dissolved in ethanol solution, which is beneficial to the hydrophobic system, and the increased lipophilicity of DHM derivatives reduced the decrease in antioxidant activity of DHM caused by losing the hydroxyl group.

### 3.2. Antioxidant Activity in the Sunflower Oil System

Sunflower oil without antioxidants was used to evaluate the antioxidant activity of DHM and its derivatives. With the oxidation of sunflower oil, many oxidation products and free radicals were continuously produced due to the loss of hydrogen atoms. The content of primary oxidation products (POVs) produced in the process of oil oxidation was an important index to evaluate oil oxidation. The POVs of oil samples supplemented with different antioxidants within 21 days were determined. As shown in Figure 2A, compared with the blank oil sample without antioxidant, TBHQ, DHM, and its derivatives delayed the oxidation process of sunflower oil from 6 days. The POVs of oil samples with added DHM and its derivatives were higher than TBHQ, indicating that TBHQ had the strongest antioxidant activity in the pure oil system. At the same time, the POVs of oil samples with added DHM were lower than those of its derivatives. Similar results were obtained using Rancimat. The induction times of the blank group and oil samples treated with DHM, C2-DHM, C4-DHM, C6-DHM, C8-DHM, C12-DHM, and TBHQ were 2.41, 5.27, 4.25, 4.27, 4.47, 4.60, 2.71, and 6.22 min, respectively. Although the lipophilicities of the acylated derivatives are improved in the sunflower oil system, their antioxidant activities are reduced. This may be due to the acylation reaction replacing a H atom of active hydroxyl group on the DHM molecule, which greatly reduces their antioxidant activities in the sunflower oil system. This result is inconsistent with that of Zhu et al. [6]. In their research, the antioxidant activities of EGCG derivatives in sunflower oil were stronger than those of EGCG, which may be because the acylation reaction of EGCG replaced the hydroxyl group, which has little effect on its antioxidant activity, while the improvement of lipophilicities of EGCG derivatives makes it play a more important role in the oil system.

In addition, for DHM derivatives, when the alkyl chain increased from C2 to C8, POVs decreased and antioxidant activities increased. When the alkyl chain increased from C8 to C12, the POVs increased and the antioxidant activities decreased, as shown in Figure 2A. The results were similar to the conclusions of Oh and Shahidi [13]. In corn oil, the antioxidant activities of resveratrol derivatives increased when the alkyl chain increased from C4 to C8 but decreased when the alkyl chain continuously increased. This may be due to the increasing solubility of derivatives in oils with the increase in alkyl chain, which enhanced their antioxidant activities to a certain extent, but when the length of alkyl chain increased to a certain threshold, the contact between derivatives and oxidation products in oils was limited due to steric hindrance, which reduced their antioxidant activities. Although the antioxidant activity of DHM in the sunflower oil system was higher than that of its derivatives, its solubility in oil was low, and it was in a floating state, which affected the appearance and evaluation of oil. The antioxidant activity of C8-DHM and DHM was similar, and its solubility in oil increased, so acylated derivatives can replace DHM in pure oil systems to a great extent.

### 3.3. Antioxidant Activity in the O/W Emulsion

The accelerated oxidation of the emulsion was induced in the dark environment at 45 °C. By determining the peroxide value and TBARS value, the inhibition effect of DHM and its derivatives on the primary oxidation products and the secondary oxidation products (malondialdehyde) produced by the oxidation of the O/W emulsion were studied. As shown in Figure 2B, the peroxide values of emulsions with added DHM and C4-DHM were less than those of the emulsion with added TBHQ from Day 10, and the peroxide values of emulsions with added DHM and C4-DHM tended to be constant after 28 d. Similarly, starting from day 15, the TBARS values of emulsions with added DHM and C4-DHM were lower than those of the emulsion with added TBHQ, as shown in Figure 2C. Therefore, the antioxidant activities of DHM and C4-DHM in the O/W emulsion were higher than those of TBHQ. For DHM and its derivatives, the peroxide values showed a trend of C4-DHM < DHM < C2-DHM < C8-DHM < C6-DHM < C12-DHM (*p* < 0.05), and the TBARS values showed a trend of DHM ≈ C4-DHM < C2-DHM < C8-DHM < C6-DHM < C12-DHM (*p* < 0.05). Generally, the antioxidant activities of DHM and its derivatives in the O/W emulsion increased first and then decreased, and except for C4-DHM, the antioxidant capacities of other derivatives were weaker than those of DHM. These results suggested that the antioxidant activities of DHM derivatives in O/W emulsions were independent of their hydrophobicities, which was inconsistent with the hypothesis of the polarity paradox.

In addition, it was generally believed that the oxidation of the O/W emulsion occurred at the oil–water interface, so the distribution of antioxidants in the O/W emulsion played a vital role in its antioxidant activity [6,29]. Therefore, to ascertain the distribution of DHM and its derivatives in the O/W emulsion, their content in the water phase of the emulsion was determined. It was found that the content in the water phase of the emulsion decreased with alkyl chain extension (Figure 2D). Combined with the analysis of antioxidant capacity, when the alkyl chain was less than C4, the corresponding antioxidant was mainly distributed in water, so the antioxidant ability was weak. When the C4 was used as an alkyl chain, DHM butyl ester was closer to the oil–water interface, so it was easier to prevent oil oxidation. When the longer alkyl chain was used, the corresponding antioxidant entered the oil phase, making the oil easier to contact with oxygen and oxidize.

### 3.4. Cellular Antioxidant Activity

By analyzing the DPPH and ABTS radical scavenging ability and the antioxidant activity in the food models, it was found that DHM and its derivatives have different antioxidant activities in different reaction media. However, these methods cannot accurately evaluate the antioxidant activities of DHM and its derivatives in biological systems. Compared with these methods, CAA tests, which can evaluate the antioxidation of substances at the cellular level, can reflect the activities of DHM and its derivatives in biological system. Some studies have shown that the antioxidant activities of bioactive substances inside and outside cells are different [19], so it is more important to assess the antioxidant activity of DHM and its derivatives by a cell model.

According to previous studies, many substances promote cell proliferation with low concentrations and inhibit cell proliferation with high concentrations. Therefore, before the CAA tests, the cytotoxicities of DHM and its derivatives were determined to reduce the toxic effect of them on cells due to the high concentration. Through cytotoxicity analysis (Figure 3), when the sample concentration was ≤240 μmol/L, the cell survival rate was >90%. The cell survival rate was <90% when the sample concentration was 480 μmol/L. Therefore, follow-up experiments were carried out with sample concentrations within 240 μmol/L to reduce the impact on the results produced by the cytotoxicity of DHM and its derivatives on cells.

The logarithmic curve of the CAA values and the concentration of DHM and its derivatives are shown in Figure 4. R^2^ was greater than 0.9, showing a good dose–response relationship, which proved the effectiveness of the L-02 cell model in evaluating the antioxidant activities of DHM and its acylated derivatives. When the CAA value was 50, the concentration for 50% of maximal effect (EC_50_) of the sample could be calculated through the logarithmic curve. Table 1 shows that the EC_50_ values of DHM and its derivatives first decreased and then increased with the growth of the alkyl chain. It showed a trend of C12-DHM > DHM > C2-DHM > C4-DHM ≈ C6-DHM > C8-DHM (*p* < 0.05). These values were all greater than the EC_50_ value of quercetin. The results indicated that the antioxidant activities of DHM and its derivatives in L-02 cells increased first and then decreased with the growth of the alkyl chain. C8-DHM had the best antioxidant activity. Bayrasy et al. [30] studied the ability of rosmarinic acid and its esters to reduce the level of intracellular reactive oxygen species (ROS). The antioxidant activity also showed a parabolic relationship with the length of alkyl chain, and the ester with medium chain length (C10) had the best antioxidant effect. It suggested that with the increase in alkyl chain, the lipophilicities of DHM derivatives increased, making it easier to approach the cell membrane, combine with the cell membrane, enter the cell, and exert its activity [19]. However, when the alkyl chain reached a certain length, the derivative would form aggregates outside the cell, which made it difficult to cross the cell membrane, and the antioxidant activity was weakened [30].

The results of cell with and without washing using PBS can evaluate the absorption of DHM and its derivatives in cells and the tightness of its binding with the cell membrane [24,31]. Therefore, after incubation with samples, the cell absorption and binding to the cell membrane were compared by measuring the antioxidant activity with and without PBS cleaning. From Table 1, the EC_50_ values of DHM and C2-DHM with washing were greater than those without washing (*p* < 0.05). C4-DHM, C6-DHM, and C8-DHM were the same as quercetin, which was not significantly different in EC_50_ values between washing and no washing (*p* > 0.05). The EC_50_ value of C12-DHM after washing was lower than that of no washing (*p* < 0.05). Washing with PBS after adding samples for incubation would wash off extracellular samples and samples loosely bound to the cell membrane, so the EC_50_ values of samples with weak lipophilicity increased and their antioxidant activity decreased.

## 4. Conclusions

In this study, the difference in antioxidant activities between DHM and its acylated derivatives with different carbon chain lengths was studied, and the relationship between lipophilicity and antioxidant activity in different reaction systems was discussed. In polar medium, except for C12-DHM, the scavenging abilities of the other DHM derivatives for ABTS and DPPH radicals were close to those of DHM, and the scavenging abilities of C12-DHM decreased significantly. In sunflower oil, the antioxidant activity of DHM was higher than that of its derivatives. In O/W emulsion and L-02 cell models, the relationship between lipophilicity and antioxidant activity was not linear. Among them, C4-DHM showed good antioxidant activity in O/W emulsion, while C8-DHM showed better antioxidant activity in the cell model. These results showed that acylation-modified DHM with improved lipophilicity has great application prospects in the food industry. Moreover, the relationship between the lipophilicity and antioxidant activity of polyphenols was different in different evaluation methods. DHM derivatives with different lipophilicities exhibit various antioxidant capacities in different application systems. Therefore, it is necessary to synthesize specific DHM derivatives according to the application model, which is also the focus of further research.

## Figures and Tables

**Figure 1 foods-12-01986-f001:**
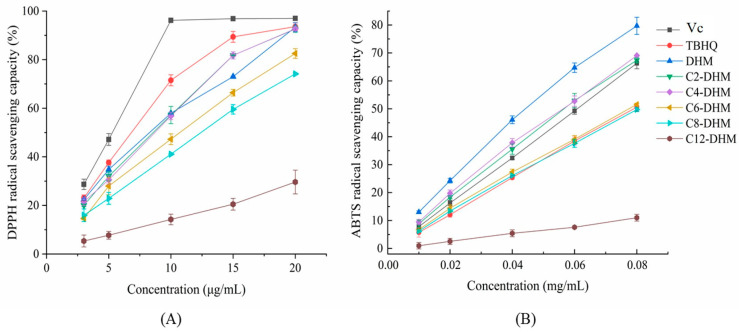
Antioxidant activities of DHM, DHM derivatives, Vc, and TBHQ in chemical media: (**A**) DPPH radical scavenging capacity; (**B**) ABTS radical scavenging capacity.

**Figure 2 foods-12-01986-f002:**
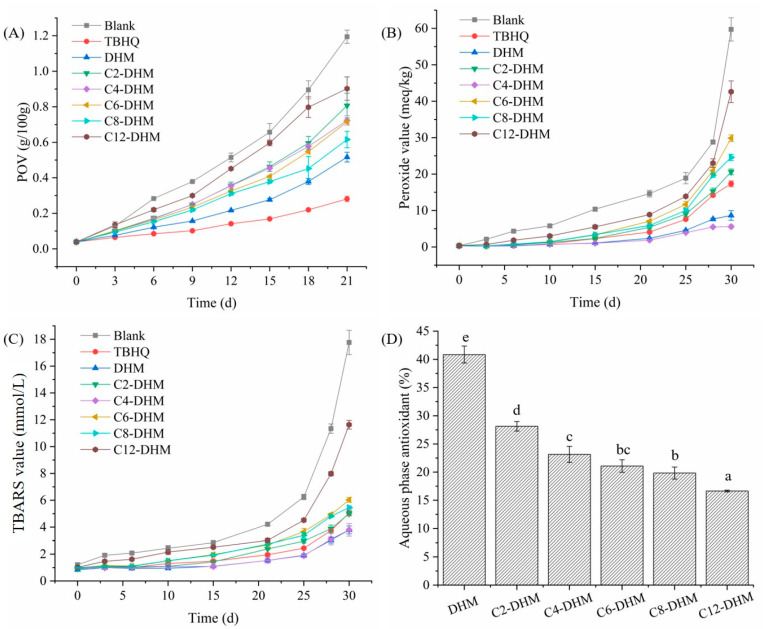
Oxidation stabilities of antioxidants in lipid-based food models: (**A**) POV in sunflower oil, (**B**) peroxide value in emulsion, (**C**) TBARS value in emulsion, and (**D**) antioxidant partitioning in aqueous phase of emulsion (different letters indicate significant difference in the same column (*p* < 0.05)).

**Figure 3 foods-12-01986-f003:**
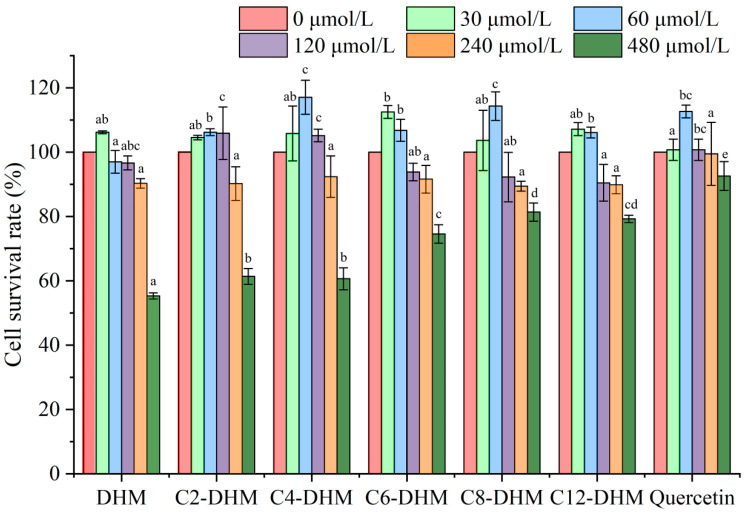
Cytotoxicity of DHM and its acylated derivatives toward L-02 cells (when the concentration of sample is the same, different letters indicate significant difference between different samples (*p* < 0.05)).

**Figure 4 foods-12-01986-f004:**
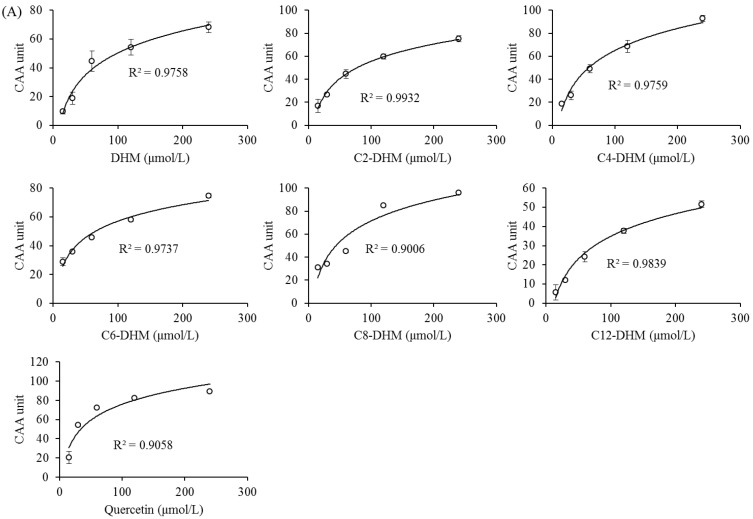
Dose–response curves for inhibition by DHM, DHM derivatives, and quercetin: (**A**) the cells were treated with no PBS wash between the antioxidant and ABAP treatments; (**B**) the cells were treated with PBS wash between the antioxidant and ABAP treatments.

**Table 1 foods-12-01986-t001:** IC_50_ values of antioxidants in DPPH and ABTS radical scavenging tests and the EC_50_ values of antioxidants in CAA test.

Sample	IC_50_		EC_50_ (μmol/L)	
	DPPH (μg/mL)	ABTS (μg/mL)	CAA Unit	
			No Wash	Wash
Vc	4.23 ± 1.38 ^a^	61.02 ± 2.15 ^a^	-	-
TBHQ	7.55 ± 0.18 ^ab^	79.34 ± 3.37 ^a^	-	-
DHM	9.04 ± 0.09 ^bc^	46.51 ± 2.19 ^a^	90.17 ± 5.02 ^e^	226.26 ± 4.67 ^e^
C2-DHM	9.07 ± 0.10 ^bc^	58.29 ± 2.20 ^a^	77.79 ± 3.81 ^d^	171.62 ± 8.84 ^d^
C4-DHM	9.08 ± 0.25 ^bc^	56.30 ± 1.03 ^a^	60.81 ± 6.23 ^c^	60.70 ± 0.64 ^b^
C6-DHM	11.17 ± 0.04 ^bc^	77.22 ± 1.46 ^a^	65.24 ± 3.00 ^c^	52.91 ± 2.69 ^b^
C8-DHM	12.69 ± 0.09 ^c^	80.15 ± 0.42 ^a^	45.43 ± 0.74 ^b^	35.14 ± 0.29 ^a^
C12-DHM	35.93 ± 5.63 ^d^	360.69 ± 50.13 ^b^	233.96 ± 9.69 ^f^	122.44 ± 9.48 ^c^
Quercetin	-	-	33.91 ± 3.67 ^a^	29.03 ± 0.36 ^a^

Significant differences (*p* < 0.05) between the samples are marked as different letters.

## Data Availability

The data presented in this study are available on request from the corresponding author. The data are not publicly available due to being used in other future works.

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
