# Peer review of "Antioxidant Activities of Dihydromyricetin Derivatives with Different Acyl Donor Chain Lengths Synthetized by Lipozyme TL IM"

_foods, 2023, doi:10.3390/foods12101986_

Round 1

Reviewer 1 Report

The article is interesting, however some comments and recommendations are made to increase the quality of the article.

Lines 116-119. In what proportions were all these compounds added?

Lines 120-123. What were the conditions used in these techniques? The chromatograms of these techniques should be attached as supplementary material.

Figure 1. What does the 1'' mean? Is this not a C of the acyl group?. Why not place the C and indicate that it is 1´´?

Lines 163-154. What amounts of sample and sunflower oil were used?

Line 165. POV was measured every 3 days, but how long was the final sampling time?

Line 184. what pH was used in the phosphate buffer?

Table 1. Why don't the authors use the units micrograms per milliliter instead of milligrams per milliliter? thus the comparison between both radicals would be better.

Lines 313-315. Why does this happen? What antioxidant mechanisms are used by both DPPH and ABTS? To which compounds are one and the other more related? Discuss more about it.

Lines 376-377. Why ?

Lines 457-472. Conclusions.  The conclusions seem like a summary of results. What is the main objective of the authors, to increase the lipid solubility of the flavonoid or to increase its antioxidant capacity in food or human systems?

Reviewer 2 Report

The manuscript titled “Antioxidant activities of dihydromyricetin derivatives with different acyl donor chains lengths synthesized by lipozyme TL IM” shows novel information regarding the antioxidant activity of dihydromyricetin derivatives measured through several antioxidant methods. The manuscript is well-written, scientifically sound, and fits within Foods’ scope. The manuscript has a high similarity with previously published manuscripts such as:

Wang et al. (2021). https://doi.org/10.1016/j.foodchem.2021.129904

Zhu et al. (2022). https://doi.org/10.3390/pr10071368

Minor Revisions are suggested for this manuscript.

Abstract

1.     The authors should give more background about the importance of dihydromyricetin, its relationship to foods, and why acetylation is needed.

Introduction

2.     Line 44: Please correct: “(…) exists in Garcinia, Vitis, and Myricaceae plants.

3.     Line 47: “protecting the liver” is a very general description. Please specify.

4.     Figure 1 is unnecessary and could be more suitable as a Supplementary File.

Materials and methods

5.     Line 100: Enzyme activity should be indicated.

6.     Line 132: Did the authors screen their results against in silico predictions?

7.     Line 143: What antioxidant control did the authors use? Same comment for ABTS. Did the authors follow a standard curve? Please specify.

8.     Line 192: Please express centrifugation units in gravity (or rcf) as rpm are centrifuge specific. Please double-check this throughout the manuscript (for example Line 203).

9.     Line 230: Is it the same assay as Intracellular Reactive Oxygen Species?

10.  Line 236: Please spell “ABAP.”

11.  Section 2.8: What statistical methods did the authors use? This is missing.

Results and discussion

12.  Fig. 2: Statistical analysis is missing. Same comment for Fig. 3A, B, and C.

13.  Line 404-405: “Positive” or “negative” are highly subjective. Please find more suitable expressions.

14.  Fig. 5: What is CAA (y-axis)? Why did the authors use quercetin? This is not clear.

Reviewer 3 Report

Lines 44-45: the expression "and its content in Ampelopsis grossedentata can reach up to 30%" must be reconsidered/reworded because it is understood that 30% of the amount of the plant is represented by dihydromyricetin.

Lines 71, 77, 78: "cut-off effect" or cut-off effect or "cut-off" effect!!!!!

The part of synthesis and characterization of the synthesized compounds I believe must be completed with the synthesis method, even if it is a method taken from the literature. The method must also describe the separation and purification of the compounds, their characterization, especially the NMR and MS spectral data, to confirm the structure. If the authors do not confirm the structures of the synthesized compounds, the results of the biological experiments have no value.

Line 165: 3 d represents tree days? The same question for other similar abbreviations in the manuscript (lines 322, 324).

Lines 175, 176: I suggest presenting s and m in bold or italics.

The experimental methods are not described explicitly enough, for example: "Then, one part was washed with PBS, the other part was not washed, and 100 μL of ABAP (600 μmol/L) was added to each well."

Line 268: the acyl residue is not a donor, but an electron attractor, due to the carbonyl group. The expression is also not suitable for the fact that the acyl residue does not replace the hydroxyl group but replaces the hydrogen of hydroxyl group. The same observation for the expression "acylation reaction replacing an active hydroxyl group on the DHM molecule" from line 333.

Fig. 4: I recommend using different colors, for easy examination by the reader. The authors must explain/rephrase for the reader the expression "(different letters indicate significant difference of different samples at the same concentration".

Round 2

Reviewer 1 Report

Los autores mejoraron la calidad del artículo y se consideraron las recomendaciones. Por lo tanto, el artículo ahora puede ser publicable.